# Complement and Cancer—A Dysfunctional Relationship?

**DOI:** 10.3390/antib9040061

**Published:** 2020-11-05

**Authors:** Joshua M. Thurman, Jennifer Laskowski, Raphael A. Nemenoff

**Affiliations:** Division of Nephrology and Hypertension, University of Colorado Anschutz Medical Campus, Aurora, CO 80045, USA; Jennifer.laskowski@cuanschutz.edu (J.L.); raphael.nemenoff@cuanschutz.edu (R.A.N.)

**Keywords:** complement, cancer, immunity, myeloid cells, therapeutics

## Abstract

Although it was long believed that the complement system helps the body to identify and remove transformed cells, it is now clear that complement activation contributes to carcinogenesis and can also help tumors to escape immune-elimination. Complement is activated by several different mechanisms in various types of cancer, and complement activation fragments have multiple different downstream effects on cancer cells and throughout the tumor microenvironment. Thus, the role of complement activation in tumor biology may vary among different types of cancer and over time within a single tumor. In multiple different pre-clinical models, however, complement activation has been shown to recruit immunosuppressive myeloid cells into the tumor microenvironment. These cells, in turn, suppress anti-tumor T cell immunity, enabling the tumor to grow. Based on extensive pre-clinical work, therapeutic complement inhibitors hold great promise as a new class of immunotherapy. A greater understanding of the role of complement in tumor biology will improve our ability to identify those patients most likely to benefit from this treatment and to rationally combine complement inhibitors with other cancer therapies.

## 1. Introduction

It was long assumed that the complement cascade contributes to the immunosurveillance of cancers, helping the body to recognize and eliminate transformed cells. In 2008, however, Markiewski and colleagues reported that mice with targeted deletion of the genes for C3 or C4 are protected from cancer in an implantation model [1]. This landmark study revealed that complement activation can promote tumor growth in some settings. Since then, studies from many different research groups have confirmed and expanded on these findings. It is now clear that the complement cascade is activated in many tumors, and that this component of the innate immune system plays a complex role in carcinogenesis and anti-tumor immunity. The mechanisms of complement activation seem to vary among different types of cancer, and the cancer cells themselves often play an active role in modulating complement activation within the tumor microenvironment (TME). For example, various types of cancer express proteins that both activate and inhibit the complement cascade within the TME.

## 2. How Does Carcinogenesis Occur?

Cancer is a disease of dysregulated growth. While our knowledge of cancer dates back centuries, it has been more challenging to actually define the critical properties of cancer. Starting in the 1980s, studies identified several somatic mutations as critical to the disease. These included activating mutations in drivers of proliferation, designated as oncogenes, and loss of function mutations in tumor suppressor genes. These mutations result in the development of the “transformed” phenotype. Transformed cells acquire new features such as the loss of contact inhibition and the ability to grow in suspension. The biological consequences of these mutations were formally characterized as the “Hallmarks of Cancer” in a seminal review by Hanahan and Weinberg in 2000 [2]. However, solid tumors originate in specific organs and are surrounded by a variety of non-transformed cells. The surrounding cell populations and stroma have been designated as the TME. The TME includes vascular cells, inflammatory and immune cells, fibroblasts, and extracellular matrix. While the initial focus on the transformed epithelial cell did not consider the TME as a driver of cancer progression, it has become apparent that the interactions between cancer cells and the TME are critical. Studies performed in the first decade of this century identified important features of the TME that regulate anti-tumor immunity and cancer metabolism and define additional “Hallmarks” of cancer [3].

A current view of cancer development needs to take into account the complexity of the interactions between cancer cells and the TME, as well as how these interactions change in a spatiotemporal fashion. Thus, epithelial cells undergo initial somatic mutations resulting in activation of oncogenic signaling or loss of tumor suppressor function. This results in increased “fitness” of these cells, giving them a survival advantage. Additional mutations occur which lead to improved cell-autonomous fitness and/or altered interactions with the surrounding TME. Thus, targeting these interactions therapeutically has become a major focus of research. 

Additional complexity in this setting is the degree of heterogeneity observed in human tumors. This is reflected by differences in mutational status and metabolic qualities of cancer cells within the same tumor. This variability makes the development of novel therapeutic approaches particularly challenging. Developing rational combinations of therapeutic agents to target this heterogeneity is often limited by the fact that most preclinical models of cancer fail to recapitulate these critical features of human disease.

## 3. Cancer, Inflammation, and Immunity

The immune system has a complex relationship with carcinogenesis. Chronic inflammation is strongly linked with the risk for many cancers, and it is generally associated with the promotion of tumor progression and metastasis. Cancer-causing inflammation can be produced by infections [4,5], environmental irritants [6,7], and autoimmune diseases [8]. For example, chronic viral infections, such as the human papillomavirus, are associated with the development of head and neck cancers. Chronic pancreatitis, hepatitis, and inflammatory bowel disease are strongly predictive of pancreatic, hepatocellular, and colon cancer, respectively. Perhaps most notably, cigarette smoking leads to the development of chronic lung inflammation and chronic obstructive pulmonary disease (COPD), and it is a major contributor to lung cancer. The immune response in these diverse settings probably contributes to genomic instability, cellular proliferation, and remodeling in the target tissues. Tissue-specific mechanisms are also undoubtedly important, however, as some organs are particularly susceptible to inflammation-associated cancers. Inflammatory changes also occur within the TME of cancers after they have formed, even in tumors for which inflammation is not an initial predisposing factor. As a result, essentially all tumors engage with the immune system as they develop and grow.

Inflammatory responses within tumors induce the recruitment of myeloid cells, including neutrophils and macrophages. Myeloid cells are initially recruited to the site of tumors as a result of specific molecules produced by the cancer cells, including cytokines, growth factors, and other molecules that attract myeloid cells and modulate their phenotype [9,10]. Macrophages are the most abundant leukocyte subtype in the TME, and they continually infiltrate the tumor [11]. This process appears to depend on the trafficking of monocytes from the bone marrow in response to specific chemokines produced by the cancer cells, such as (C-C motif) ligand 2 (CCL2). These innate immune cells are critical to the general response to injury and acute infection and act to eliminate infections and promote healing. Similar to what is seen during infection, the initial phenotype of myeloid cells in the TME is generally proinflammatory, designated as M1 macrophages and N1 neutrophils. The phenotypes of these cells become altered in the setting of chronic inflammation, however. Similarly, cross-talk between cancer cells and myeloid cells eventually results in modulation to alternatively activated phenotypes, designated as M2 or N2 [12]. These cells then actually promote tumor progression through the production of growth factors and proangiogenic cytokines which, in turn, signal back to the cancer cells. While this model is definitely an oversimplification, it serves as a framework for understanding the interactions between the cancer cells and these innate immune cell populations.

Cancer cells express mutated or aberrantly expressed proteins on their cell surface. These so-called neoantigens can be recognized by the adaptive immune system (CD8 and CD4 T cells) potentially leading to immune elimination. However, cancer cells can evade immune attack through multiple mechanisms. A model for this, designated as immunoediting, was proposed several years ago by Schreiber and colleagues [13]. This model comprises three stages of interaction with the immune system. Initially, tumor cells can be eliminated by the immune system; however, responses of the tumor lead to an equilibrium where the tumors are not eliminated but held in check by the immune system. Eventually, through the activation of additional cancer cell-autonomous and non-autonomous mechanisms, the cancers evade immune attack and escape. 

Numerous mechanisms control these events. Cancer cells can develop additional mutations that target antigen-presentation pathways, thus becoming invisible to the immune system. Alternatively, myeloid cells within the TME can undergo phenotypic modulation, acquiring anti-inflammatory properties [14]. The cells can then inhibit CD4 T cells and block CD8 T cell-mediated killing. Finally, a variety of pathways that regulate the immune system under non-cancerous conditions can be co-opted by cancer cells to block T cell function. Prominent among these are immune checkpoints expressed on the surface of cancer cells and other cell types which block T cell activation and lead to an “exhausted” T cell phenotype [15]. Targeting these pathways through the use of checkpoint inhibitors can result in T cell reactivation and tumor regression. These agents have been approved in a variety of cancers and have revolutionized the treatment of many types of cancer [16,17]. For several reasons that are only partially understood, however, the majority of patients either do not respond to these agents or develop resistance.

A major challenge, therefore, is the development of rational combinations of therapies that will increase the number of responders as well as the duration of the response to treatment. In particular, targeting inflammation and the cross-talk between innate and adaptive immunity could enhance the ability of the immune system to eliminate tumors.

## 4. The Role of the Complement System in Carcinogenesis

The complement system is a cascade of proteins that form part of the innate immune system. Complement factors circulate as inactive precursor proteins (zymogens) that are cleaved and activated by three different pathways: the classical, lectin, and alternative [18]. Activation of these pathways generates soluble fragments (C3a and C5a, or the “anaphylatoxins”) and also covalently fixes protein fragments (C3 and C4 fragments) on the surface of target cells. Immune cells express specific receptors for the anaphylatoxins and C3 fragments and this interaction links complement activation fragments with modulation of immune cell function. Consequently, complement activation has strong effects on innate and adaptive immunity. Full complement activation also generates multimeric complexes that form pores in target cells. In the literature, this is variably referred to as the membrane attack complex (MAC), the terminal complement complex, or C5b-9.

Complement activation is traditionally regarded as proceeding through three different pathways: the classical, lectin, and alternative pathways. The classical and lectin pathways are activated by specific proteins that activate these cascades after binding to target ligands. IgG and IgM bound to target antigens activate the classical pathway, and mannose-binding lectin bound to target sugars activates the lectin pathway [19]. The alternative pathway is spontaneously activated in plasma through a process called “tickover”. It is also secondarily activated by the classical and lectin pathways, thereby amplifying their effects.

### 4.1. Complement Activation as a Cause of Cancer

Carcinogenesis involves several steps, including the acquisition of a series of mutations that give a cell growth or survival advantages (“initiation”) and proliferation and/or decreased death of the transformed cells (“promotion”) [20]. The immune system is integrally involved with both of these stages of carcinogenesis. In most inflammatory settings, complement activation occurs as part of a broader immune response, so it is difficult to distinguish the effects of complement activation from those of other components of the immune system. Nevertheless, we recently published a study showing that chronic complement activation in the liver causes cancer [21]. Mice with targeted deletion of the gene for factor H, a key complement regulatory protein, have spontaneous complement activation within the liver [21]. As the mice aged, they developed hepatocellular carcinoma at a greater rate than control mice, whereas complement deficient mice did not. Although the experiments did not determine which stage(s) of tumorigenesis complement directly affects, it did demonstrate that, at least in the liver, chronic complement activation is sufficient to cause tumor formation.

There is a rationale for suspecting that the complement system plays a part in tumor initiation (Figure 1). Activated macrophages [22] and neutrophils [23] produce reactive oxygen species and reactive nitrogen intermediates, molecules that can cause DNA mutations [24]. C3a and C5a are strongly chemotactic for these cells and induce the cells to undergo oxidative burst. Thus, complement activation within inflamed tissues is probably integrally involved in these initiation events [25]. In support of this, Bonavita and colleagues studied the role of pentraxin-3 (PTX3) in models of chemically induced sarcomas and skin carcinomas [26]. PTX3 is expressed by many types of cancers, and it can bind to circulating factor H, tethering this inhibitor to cells and suppressing complement activation within the TME. The authors found that PTX3-deficient mice were susceptible to chemically induced sarcomas and skin carcinomas, presumably because inadequate control of complement activation increases carcinogenesis in affected tissues. Furthermore, complement deficiency reversed this tumorigenic effect. The authors also noted that PTX3-deficient mice developed a greater degree of DNA damage and a higher number of P53 mutations at early timepoints compared to wild-type mice, further supporting the conclusion that complement-mediated inflammation contributes to genomic instability and cancer initiation.

Once cells have acquired cancer-causing mutations, the promotion of the nascent tumor requires the proliferation and survival of the transformed cells [27,28]. This involves pro-mitotic signals as well as angiogenic signals to support the expanding tumor, processes that have been linked to inflammatory cytokines. The complement system may indirectly contribute to this process by inducing the production of cytokines and chemokines [29]. Furthermore, in vitro experiments have also shown that C3a and/or C5a directly stimulate cancer cell proliferation [30,31,32]. Consistent with these mitogenic effects on cancer cells, studies in non-cancer models have also shown that C3a and C5a induce survival and proliferation of cells during tissue regeneration [33,34,35]. In addition to C3a and C5a, the insertion of MAC in tumor cell membranes can stimulate cell proliferation [36] and induce chemokine and metalloprotease production by the cells [37]. These findings support a role for complement activation in tumor promotion.

Regions of hypoxia develop in solid tumors as they expand, and angiogenesis becomes essential for all tumor growth. Complement is frequently activated within ischemic tissues, and hypoxia induces some cell types, including non-small cell lung cancer cells, to decrease expression of complement regulatory proteins in vitro [38]. Thus, complement may be preferentially activated in hypoxic regions of a tumor. Complement activation has been linked with angiogenesis [39], raising the possibility that the complement activation provides a link between tissue hypoxia and the production of angiogenic signals. A comprehensive analysis of the mutational landscape in tumors revealed that there is cross-talk between the complement system and hypoxic signals in some cancers [40]. Nevertheless, in one study of non-small cell lung cancer, the C5a blockade did not have a detectable effect on vascular density within tumors [41].

### 4.2. Complement as a Mechanism of Immune Evasion

The study by Markiewski and colleagues used an implantation model of cancer [1]. Because the mice were injected with cells that were already transformed, the study implicated complement in the growth of existing tumors rather than in cancer initiation per se. In that study, the investigators showed that the generation of C5a attracted myeloid-derived suppressor cells (MDSCs) to the tumor, which reduced the anti-tumor response of CD8 T cells. Complement activation in that model, therefore, is analogous to an immune checkpoint insofar as it suppresses the immune-elimination of cancers (Figure 1). Over the subsequent 12 years, additional pre-clinical studies have reported a similar role for complement activation in promoting the growth of other types of cancer, including lung [42], squamous cell [43], melanoma [44], colon [45], and ovarian cancer [30]. Many other effects of complement on cancer cells and the TME have been identified, but the pattern of inducing myeloid cells to suppress the adaptive immune system has been generalizable across tumor types. 

Studies in multiple different types of cancer have also confirmed that the pro-tumorigenic effects of complement are mediated by immunosuppressive myeloid cells, such as MDSCs [1,41,46]. The anaphylatoxins generally have pro-inflammatory effects, but studies in solid organ transplant models have shown that C5a can also attract immunosuppressive myeloid cells into tissues [47]. It is not clear whether it is the context or the duration of complement signaling that determines whether the net effect is pro or anti-inflammatory. Interestingly, studies have revealed that both C3a and C5a contribute to the immunosuppressive effect of complement within tumors [42,44]. Furthermore, C3 produced by stellate cells in the liver affects dendritic cell maturation and attracts MDSCs, facilitating the growth of hepatocellular carcinoma [48]. Another study of stellate cells indicated that this immunomodulatory effect is caused by the iC3b fragment [49]. Thus, multiple different complement activation fragments seem to interact with myeloid cells in the TME, helping the tumor to evade immune-elimination. 

Complement activation may also play a role in protective anti-tumor immune responses. For example, it has been shown to be involved in the formation of tertiary lymphoid structures (TLS), which are structures consisting of B and T cells as well as dendritic cells. Interactions between these cells result in strong immune activation, and the presence of these structures is associated with a good response to immunotherapy in multiple cancers [50]. Complement activation in the setting of chemotherapy has been shown to promote a subset of B cells that regulate the formation of these structures [51].

Distinct from systemic production of complement, recent studies have identified key functional roles for intracellular production of complement, specifically in human CD4+ T cells [52,53]. Activation of T cells is associated with translocation of intracellular C3a and C3b to the cell surface, where these fragments regulate T cell activation and metabolism through the engagement of the C3a receptor (C3aR) and the regulatory protein CD46, respectively. These data would indicate that in the setting of cancer progression complement activation in cancer cells and T cells play opposing roles, with cancer cell complement mediating immunosuppression and T cell complement mediating T cell activation. Further research is required to dissect out the relevant importance of these pathways in specific malignancies. One complication in studying the T cell intracellular complement pathway is the fact that CD46 is not expressed in mice, making it difficult to assess this pathway in preclinical models.

### 4.3. Complement Activation and Metastatic Spread of Cancer

Beyond its effects on carcinogenesis and tumor growth, there is also experimental evidence that the complement cascade increases the invasiveness and metastatic potential of cancer [recently reviewed in [41]]. Complement activation has been linked to the metastatic potential of colon cancer [54] as well as leptomeningeal metastases of cancer cells [55]. The effect of complement fragments on metastases may be due to their direct effects on cancer cells as well as their effect on tissue remodeling in the TME and metastatic niche. C5a, for example, directly induces epithelial-mesenchymal transformation (EMT) of hepatocellular carcinoma [56] and gastric cancer cells [57], and can induce expression of metalloproteinases [58]. Conversely, the C5a receptor blockade reduces some hallmarks of EMT [59]. Work in animal models has also shown that the C5a blockade reduces metastasis of colon and lung cancers by regulating immune responses and the premetastatic niche [46,54].

## 5. Cancer as a Cause of Complement Activation

Complement activation contributes to the development of cancer by the mechanisms discussed above, but the reverse is true too: tumors actively promote complement activation within the TME (Figure 1). Inflammatory cells and molecules are present within the TME of essentially all cancers, regardless of their tissue of origin [60]. The adaptive immune response to cancer cells is a function of tumor immunogenicity, which is determined by both the antigenic burden in the cancer and the host immune system [61]. Inflammation can cause DNA damage, but DNA damage can cause the production of mutant proteins that are antigenic and elicit an immune response [62]. Cancer cells that cannot evade the immune system will be eliminated, so the cancer must either stop producing the target antigen, downregulate MHC, or suppress the immune response to the tumor antigen through immunoediting [63,64]. Although the anti-tumor immunity is primarily executed by CD4 T cells, CD8 T cells, and natural killer (NK) T cells, B cells are also seen within the TME of some tumors, and patients often develop IgG that is specifically reactive with tumor antigens [65,66]. This can serve as a link between the adaptive immune response to a tumor with complement activation within the TME. 

There is evidence of classical pathway activation in several animal models of cancer. The growth of TC-1 tumors was significantly reduced in mice with targeted deletion of C4 (classical and lectin pathway deficient), but tumor growth was not affected by the deletion of factor B (alternative pathway deficient) [1]. Similarly, studies in non-small cell lung cancer showed that IgM and C4 are deposited in both a mouse model and in human samples, pointing to the involvement of the classical pathway in this type of cancer [42]. Furthermore, tumor growth was not affected by factor B deficiency in this model. A study using the TC-1 cell line also found that C1q was required for tumor growth, further supporting the involvement of the classical pathway [67]. C1q deposits are seen in human lung, colon, breast, and pancreatic adenocarcinoma, as well as melanoma [68]. Because patients so frequently develop antibodies against tumor antigens, it is logical that classical pathway activation would be a frequent occurrence. In a mouse model of melanoma, however, investigators have also shown that C1q contributes to tumor growth in a complement-independent fashion [68]. Studies in models of systemic lupus erythematosus, an autoimmune disease, revealed that C1q directly affects the CD8 T cell response to antigens independent of the classical pathway [69]. In that study, C1q-deficient CD8 T cells displayed greater reactivity against autoantigens and viruses. Such an effect could also potentially modulate anti-tumor immunity. 

In addition to the adaptive B cell response to foreign antigens, natural antibodies are produced by B-1 cells in the absence of an antigenic stimulus and often react with carbohydrate epitopes [70]. Tumor cells frequently display abnormal post-translational modifications, including altered glycosylation patterns. Natural antibodies can bind to carbohydrate epitopes, and they have been found to react to glycans displayed on the surface of cancer cells [71]. Interestingly, a hybridoma generated from a patient with signet-ring cell carcinoma of the stomach reacts specifically with an N-linked carbohydrate on CD55 [72]. CD55 is a cell surface complement regulatory protein that protects cancer cells from complement-mediated lysis and contributes to chemoresistance of the cells [51]. Antibodies that bind to neo-epitopes displayed on the surface of CD55 could activate the classical pathway on the target cell while simultaneously impairing complement regulation by CD55 on the cell surface.

Although the requirement of C4 expression for tumor growth could also reflect a role for the lectin pathway in the TC-1 model, there is less published evidence for specific involvement of this pathway in cancer models. However, one recent study found that certain species of gut fungi can migrate to the pancreas and foster the development of pancreatic ductal adenocarcinoma [73]. MBL was required for tumor growth in this model, indicating that the lectin pathway activation was probably involved. This study also showed that signaling through the C3a receptor was necessary for tumor growth. 

Although several studies have found that an intact alternative pathway is not necessary for tumor growth [1,42], it is notable that alternative pathway proteins are expressed by most tumors [74]. The alternative pathway also amplifies complement activation that is initiated through the other pathways, so this pathway may contribute to tumor growth by increasing the overall magnitude and duration of complement activation, even if it is not essential for the reaction. Our recent study using factor H deficient mice showed that chronic alternative pathway activation in the liver is sufficient to cause HCC [21]. However, it is possible that the liver—as the source of the alternative proteins factor B and C3—may be uniquely susceptible to alternative pathway-mediated injury.

It is striking that cancer cells and other cells within the TME synthesize complement proteins [74]. Several inflammatory cytokines induce parenchymal cells to produce complement proteins, and the transcription factor twists basic helix-loop-helix transcription factor 1 (TWIST1) has been identified as a key regulator of the expression of C3 by cancer cells [75]. Although high levels of complement proteins are already present in plasma, studies in organ transplantation revealed the importance of locally produced proteins in activation within tissues [76]. Similarly, studies in cancer models have revealed that complement proteins produced by the cancer cells, themselves, contribute to tumor growth in vivo despite the expression of the same proteins by the host [30].

Various proteases are also able to directly activate complement proteins, potentially bypassing the conventional initiation mechanisms and convertases. The cathepsins, for example, are a family of serine proteases that are upregulated and secreted by many different types of cancer [77], and cathepsin L can cleave C3 into C3a and C3b [52]. Human melanoma cells express cathepsin L and cleavage of C3 by the protease is associated with growth and metastasis [78]. Similarly, thrombin is capable of directly cleaving complement C5 [79]. Cancer cells can, directly and indirectly, activate thrombin [80], providing a mechanism for the direct generation of the terminal complement components. Additional cancer-associated proteases capable of activating complement proteins have also been identified, including prostate-specific antigen [81].

## 6. Complement Regulatory Proteins

In addition to the many molecules that activate the complement pathways, the body also expresses a family of proteins that inhibit activation of the cascade [82]. Some soluble regulators circulate in plasma and other body fluids, as well as regulators expressed on the outer membrane of all cells. The various regulatory proteins inhibit complement activation by distinct mechanisms and at different sites within the cascade.

All cancer cells express cell surface complement regulators, and most cancer cells express more than one of the proteins. The cell surface regulators CD46, CD55, and CD59, for example, are all expressed by squamous cell cancers [83] and uveal melanomas [84]. Multiple studies have also shown that cancer cells overexpress complement regulators compared to the corresponding tissue of origin [85,86,87]. For years, the presumption was that overexpression of the complement regulatory proteins is a mechanism by which cancer cells evade complement-mediated elimination. In support of this, functional studies in which expression of the proteins is knocked down with small interfering RNA (siRNA) have confirmed that the regulatory proteins protect the cells from lysis [88]. Expression levels of the regulatory proteins have also been shown to correlate with complement activation with the tumor and with patient outcomes [89,90]. Interestingly, cancer cells frequently overexpress factor H, and factor H-like protein 1 (FHL-1), soluble proteins that are already present at high concentrations in plasma [91]. Studies have shown that factor H is expressed by ovarian, squamous, and colon cancer cells [92,93]. Knockdown of factor H production can reduce tumor cell survival, proliferation, and migration in vivo [94,95]. Although liver-derived factor H and FHL-1 is almost certainly present in the TME, production of these proteins by cancer cell may ensure that local concentrations are sufficient to protect the cells. Cancer cells can also express non-complement proteins, such as osteopontin, that bind to factor H, holding it within the TME [96].

Several studies have linked the expression of complement regulatory proteins by tumors with adverse clinical outcomes [97,98]. In patients with breast cancer, for example, the expression of CD59 was associated with lung metastasis and shorter [98]. Similarly, in cholangiocarcinoma higher expression of CD55 is associated with shorter survival [97]. Unsurprisingly, being able to control complement activation gives a cancer cell a survival advantage, but this does lead to a paradox. On the one hand, complement activation supports tumor growth in many contexts, yet cancer cells also actively express proteins that inhibit the complement system. It is noteworthy, however, that the expression of complement regulatory proteins by cancer cells is quite heterogeneous, and cells within the same tumor can express different repertoires of regulatory proteins [99]. Expression of these proteins is also dynamic, changing in response to local conditions such as hypoxia and inflammatory cytokines [100,101]. One model to account for these data would suggest that regulatory proteins protect cancer cells from the deleterious effects of complement, such as lysis through the MAC complex while allowing the pro-tumorigenic effects mediated by C3a/C5a to act on the TME leading to immunosuppression. Thus, complement activation may have opposing effects on tumor growth, and the effects may fluctuate depending on time and location within the tumor. Furthermore, while complement activation may destroy a limited number of cells within the tumor mass, the immunosuppressive effects of C3a and C5a may suppress the immune system throughout the TME (Figure 2).

## 7. Complement Inhibitors as Cancer Treatment

As outlined above, complement inhibitory drugs are effective for reducing tumor size in multiple different pre-clinical cancer models [1,41,42,44,102,103,104]. Inhibitory anti-C5 antibodies have been approved for several non-malignant diseases [105,106], and many additional complement inhibitory agents are in development [107]. Some of the new drugs block all activation, some selectively block the various activation pathways, and others target specific complement fragments and receptors. Thus, shortly there will likely be a range of therapeutic options for blocking the complement system. C3 fragments clearly play a role in tumor biology, but multiple studies have shown that C5aR antagonism is an effective cancer treatment. Several C5a antagonists are currently in clinical development for other indications, so this approach is already feasible.

If complement inhibitors are to be used to treat cancer, then it will be important to understand: (1) which types of cancer are responsive to this approach, (2) when in the course of the disease is complement inhibition most effective, and (3) which components of the complement system contribute to tumor growth. Furthermore, based on the experience with immune checkpoint inhibitors, complement inhibitory drugs will be most effective when combined with other therapies. The efficacy of including complement inhibitors in combination therapy likely depends on the treatments with which they are paired. For example, the efficacy of monoclonal antibodies often involves complement-mediated cytotoxicity. Consequently, the use of complement inhibitors could undermine the efficacy of these drugs. Indeed, some drugs impair the ability of cancer cells to regulate the complement cascade, thereby increasing complement activation on the cell surface. These agents have been used as a means of sensitizing tumors to monoclonal antibody treatments [108].

The combination of anti-complement drugs with immune checkpoint inhibitors may also represent a special case. In many studies, complement blockade has been shown to affect existing tumors via its effects on myeloid cell recruitment and polarization. This, in turn, enhances the anti-tumor effects of T cells. Complement inhibition and immune checkpoint inhibition, therefore, have similar, and possibly redundant, effects on anti-tumor immunity. However, preclinical studies also suggest that these drugs work by different mechanisms and can have additive effects on tumor size [104,109]. Indeed, a clinical trial is currently underway in which an anti-C5aR antibody is being combined with a PD-L1 antagonist for the treatment of solid tumors (NCT03665129; the type of cancer has not been disclosed). Finally, the development of novel targeted complement inhibitors might represent a therapeutic strategy to selectively regulate complement at the site of the tumor, without the adverse effects that might occur with systemic complement inhibition. Several fusion proteins have been developed which need to be tested in appropriate preclinical models [110].

## 8. Conclusions

One of the most unexpected and powerful discoveries in complement research is the critical role of this system in cancer biology. Although the full range of effects that complement fragments have on cancer initiation and tumor growth are not yet known, a frequent finding is that complement inhibition increases anti-tumor immunity and reduces tumor size. This is, in many respects, analogous to checkpoint inhibition. Many different complement inhibitors are in clinical development, and this class of drugs holds great promise for the treatment of cancer. Based on what is currently known about the role of complement in carcinogenesis, there is reason to suspect that complement inhibition can reduce the risk of tumor initiation in patients with some chronic inflammatory diseases, and it can slow the growth and spread of many types of cancer.

The optimal use of complement inhibitory drugs will entail a greater understanding of the role of complement activation in specific types of cancer, as well as the potential and limitations of combining complement inhibitory drugs with various other cancer therapies. Studies have shown that cancer cells express many different proteins that promote and inhibit complement activation, highlighting a complex interaction between cancer cells and complement proteins that may vary over time and throughout the tumor. Although disentangling all the details of this relationship will be challenging, many of the tools developed for complement research in other fields are being applied to cancer models. Furthermore, activation of the complement system generates several tissue bound and soluble biomarkers, which may provide accurate methods by which complement activation within a tumor can be monitored in the clinic.

## Figures and Tables

**Figure 1 antibodies-09-00061-f001:**
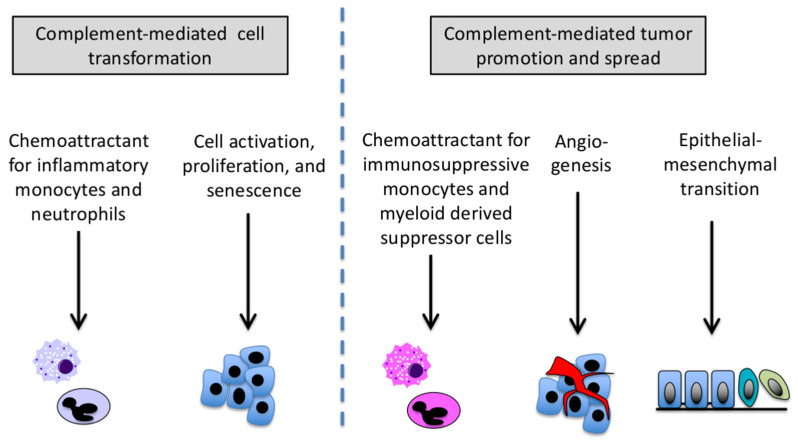
Possible roles for complement in tumor initiation and promotion. Complement activation is a frequent component of the acute inflammatory response. Complement fragments are chemotactic for myeloid cells, and also elicit the production of chemokines by epithelial cells. This contributes to inflammatory injury. Complement also directly stimulates cell activation and proliferation in target tissues. Once cancers have developed, complement fragments protect the tumor from immune-elimination by attracting immune-suppressive myeloid cells into the tumor microenvironment. Complement fragments may be involved with angiogenesis, which supports the expanding tumor mass. It may also trigger epithelial-mesenchymal transition, which contributes to tissue remodeling and tumor spread.

**Figure 2 antibodies-09-00061-f002:**
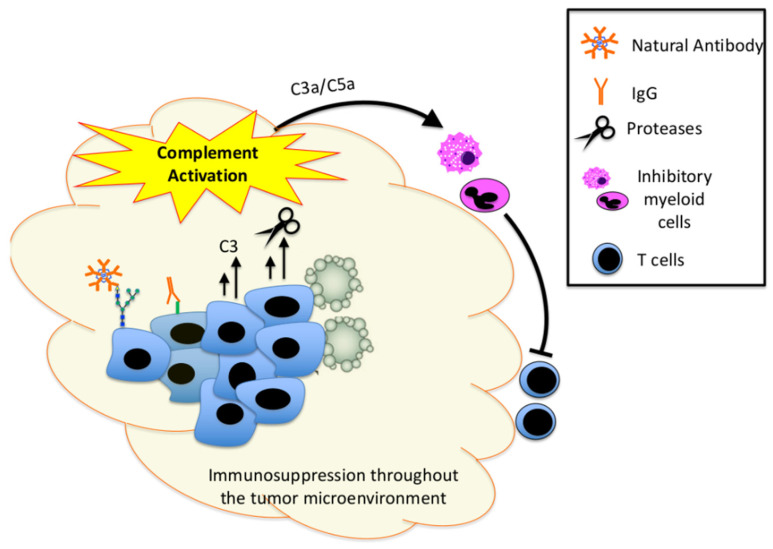
Mechanisms by which cancer cells activate complement to control the tumor microenvironment. Tumor cells actively promote complement activation by several mechanisms. The natural antibody binds to glycans on the cell surface, and IgG binds to tumor neoepitopes. Cancer cells also produced complement proteins, such as C3, which fuels local activation. Cancer cells can also release proteases, such as cathepsin L, which directly activate complement proteins. Complement activation within tumors likely causes apoptosis and necrosis of some target cells, but it also produces C3a and C5a which recruit inhibitory myeloid cells into the tumor microenvironment. These myeloid cells suppress the anti-tumor function of CD4 and CD8 T cells.

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
