# Peer review of "Complement and Cancer—A Dysfunctional Relationship?"

_2073-4468, 2020, doi:10.3390/antib9040061_

Round 1

Reviewer 1 Report

My overall impression is of a well-written, comprehensive review of the complicated interaction of the complement system with tumors and the TME. The authors do an excellent job of highlighting both the pro- and anti-tumor roles of complement and how these might be utilized to improve cancer immunotherapy in the appropriate context. A few minor corrections and suggestions are listed below.

Line 130 correct typo. "...cascade of proteins that form part.. "

Line 135 Consider including a statement to clarify how complement activation has an effect of innate and adaptive immunity. For example, "Immune cells express specific receptors.... C3 fragments and this interaction modulates immune cell function.

Line 179 instead of all tumors, consider stating solid tumors

Line 214 Consider a new paragraph and/or stating that the sentence beginning with, "Complement activation has also been shown to be..", is a pro-host response in contrast to the rest of the paragraph.

Line 222 Correct statement regarding CD4 T cells (C3a and CD46 do not interact). Intracellular C3a and C3b translocate to the cell surface and interact with their respective receptor (C3aR) or the regulatory protein CD46.

Line 234 check citation: reviewed in [38]

Line 246 Should refer to Figure 2, not Figure 1? Check placement of Figure references in ms

Consider merging Sections 5 and 6. Suggestion- Paragraph starting at Line 289 follows paragraph Line 244. Paragraph 269 follows Line 308. Paragraph at Line 279 proceeds paragraph Line 323.

Clarify alternative pathway in paragraph starting at Line 289 for non-experts

Line 309 add "in the TC-1 model" for clarity.

Author Response

We appreciate the careful review of ourmanuscript by the reviewers and editor. The authors agree with the suggestions of the reviewers and have revised the manuscript as suggested.

Line 130 correct typo. "...cascade of proteins that form part.. "

This has been done.

Line 135 Consider including a statement to clarify how complement activation has an effect of innate and adaptive immunity. For example, "Immune cells express specific receptors.... C3 fragments and this interaction modulates immune cell function.

This has been done.

Line 179 instead of all tumors, consider stating solid tumors

This has been done.

Line 214 Consider a new paragraph and/or stating that the sentence beginning with, "Complement activation has also been shown to be..", is a pro-host response in contrast to the rest of the paragraph.

This has been done.

Line 222 Correct statement regarding CD4 T cells (C3a and CD46 do not interact). Intracellular C3a and C3b translocate to the cell surface and interact with their respective receptor (C3aR) or the regulatory protein CD46.

This has been corrected.

Line 234 check citation: reviewed in [38]

The citation has been checked.

Line 246 Should refer to Figure 2, not Figure 1? Check placement of Figure references in ms

The figure callouts and placement has been checked.

We have adjusted the callouts for the figures.

Consider merging Sections 5 and 6. Suggestion- Paragraph starting at Line 289 follows paragraph Line 244. Paragraph 269 follows Line 308. Paragraph at Line 279 proceeds paragraph Line 323.

This has been done.

Clarify alternative pathway in paragraph starting at Line 289 for non-experts

This has been done.

Line 309 add "in the TC-1 model" for clarity.

This has been done.

Reviewer 2 Report

This is an excellent manuscript, describing the role of the complement system in carcinogenesis. It is original and based on the latest development in the field. It is well written and easy to follow. 

275-278: The authors discuss CD55 in cancer. The authors can mention one sentence on the role of CD55, expressed on tumors cells, which induces chemoresistance by suppressing the  complement activation, Lu et al, Cell, 2020, current ref 48. 

Author Response

We appreciate the careful review of our manuscript by the reviewers and editor. The authors agree with the suggestions of the reviewers and have revised the manuscript as suggested.

275-278: The authors discuss CD55 in cancer. The authors can mention one sentence on the role of CD55, expressed on tumors cells, which induces chemoresistance by suppressing the  complement activation, Lu et al, Cell, 2020, current ref 48. 

This has been done.

Reviewer 3 Report

In the manuscript, authors have described the paradoxical nature of complement system in cancer and its environment. The earlier assumption of complement cascade as an immunosurveillance of cancers has been questioned by number of studies reported in last decade or so.

The authors have done an excellent job in explaining the carcinogenesis and the role complement activation plays in each phase such as initiation and proliferation. The dual nature of these complement systems in cancer cells is intriguing as cancer cells express complement regulatory proteins themselves.

The review should be accepted for publication with some minor revisions.

  1. On page 3, In the third paragraph references are missing. For example, myeloid cells within TME can undergo phenotypic modulation,…… (Ref)
  2. Page 3, Line 130 - The sentence structure “The complement system is a cascade…..” is confusing.
  3. Page 9, last paragraph – More discussion on complement regulatory proteins and type of cancer cells along with their heterogeneous response would be helpful.  

Author Response

We appreciate the careful review of our manuscript by the reviewers and editor. The authors agree with the suggestions of the reviewers and have revised the manuscript as suggested.

  1. On page 3, In the third paragraph references are missing. For example, myeloid cells within TME can undergo phenotypic modulation,…… (Ref)

We have added references to this paragraph.

  1. Page 3, Line 130 - The sentence structure “The complement system is a cascade…..” is confusing.

This has been corrected.

  1. Page 9, last paragraph – More discussion on complement regulatory proteins and type of cancer cells along with their heterogeneous response would be helpful.  

We have expanded the discussion in this paragraph.